# Protein Kinase A Negatively Regulates the Acetic Acid Stress Response in *S. cerevisiae*

**DOI:** 10.3390/microorganisms12071452

**Published:** 2024-07-17

**Authors:** Natasha M. Bourgeois, Joshua J. Black, Manika Bhondeley, Zhengchang Liu

**Affiliations:** 1Department of Biological Sciences, University of New Orleans, New Orleans, LA 70148, USA; 2Center for Global Infectious Disease Research, Seattle Children’s Research Institute, Seattle, WA 98109, USA; 3Department of Molecular Biology and Genetics, Johns Hopkins University School of Medicine, Baltimore, MD 21205, USA; 4Kudo Biotechnology, 117 Kendrick Street, Needham, MA 02494, USA

**Keywords:** acetic acid stress response, protein kinase A (PKA), *S. cerevisiae*, Haa1, Ras2, Pde2, Tpk1, Tpk2, Tpk3

## Abstract

Bioethanol fermentation from lignocellulosic hydrolysates is negatively affected by the presence of acetic acid. The budding yeast *S. cerevisiae* adapts to acetic acid stress partly by activating the transcription factor, Haa1. Haa1 induces the expression of many genes, which are responsible for increased fitness in the presence of acetic acid. Here, we show that protein kinase A (PKA) is a negative regulator of Haa1-dependent gene expression under both basal and acetic acid stress conditions. Deletions of *RAS2*, encoding a positive regulator of PKA, and *PDE2*, encoding a negative regulator of PKA, lead to an increased and decreased expression of Haa1-regulated genes, respectively. Importantly, the deletion of *HAA1* largely reverses the effects of *ras2*∆. Additionally, the expression of a dominant, hyperactive *RAS2^A18V19^* mutant allele also reduces the expression of Haa1-regulated genes. We found that both *pde2*Δ and *RAS2^A18V19^* reduce cell fitness in response to acetic acid stress, while *ras2*Δ increases cellular adaptation. There are three PKA catalytic subunits in yeast, encoded by *TPK1*, *TPK2*, and *TPK3*. We show that single mutations in *TPK1* and *TPK3* lead to the increased expression of Haa1-regulated genes, while *tpk2*Δ reduces their expression. Among *tpk* double mutations, *tpk1*Δ *tpk3*Δ greatly increases the expression of Haa1-regulated genes. We found that acetic acid stress in a *tpk1*Δ *tpk3*Δ double mutant induces a flocculation phenotype, which is reversed by *haa1*Δ. Our findings reveal PKA to be a negative regulator of the acetic acid stress response and may help engineer yeast strains with increased efficiency of bioethanol fermentation.

## 1. Introduction

Weak organic acids are commonly used to preserve foods by inhibiting the growth of microorganisms. The budding yeast, *Saccharomyces cerevisiae,* has multiple regulators that counteract the inhibitory effect of weak acids on growth, including Msn2/4, Rim101, Haa1, War1, and Pdr1/3, among others (reviewed in [1]). Haa1 is specifically involved in cellular adaptation to short chain, less lipophilic weak acids, such as acetic acid and propionic acid, while War1 is involved in adapting cells to more lipophilic weak acids such as sorbic acid and benzoic acid [2,3]. Understanding the mechanisms of yeast adaptation to weak organic acids can help inform strategies to improve food preservation when microbial growth is to be limited and engineer yeast strains to increase the efficiency of industrial biofermentations where robust growth is desired.

Acetic acid is a common inhibitor in industrial fermentation processes, and the adaptive response in the presence of sublethal concentrations of acetic acid in *S. cerevisiae* has been actively researched in recent years [1,3,4,5,6,7,8,9]. In the presence of lethal concentrations of acetic acid, yeast cell death and apoptosis ensue [10,11,12]. *S. cerevisiae* has evolved a complex network of physiological and cellular responses to mitigate the effect of acetic acid. A key component of this adaptative response is the transcription factor Haa1, which activates the expression of many genes that contribute to detoxification and adaptation to acetic acid [4,6,13]. One of the primary responses is the upregulation of genes encoding an activator of the plasma membrane H^+^-ATPase, Hrk1, and membrane transporters Tpo2 and Tpo3, which help to export protons and possibly acetate to restore intracellular pH balance [3,14,15]. Other Haa1 targets, such as Spi1 and Ygp1, may contribute to the remodeling of the cell envelope and decrease the diffusion of acetic acid across the plasma membrane [3,16], while Yro2 localizes to the plasma membrane and is required for adaptation to acetic acid, but its mechanism is unclear [3,17].

Haa1 shuttles between the cytoplasm and nucleus [18] and is mostly localized in the cytoplasm of unstressed cells. Weak organic acid treatment induces its nuclear translocation and transcriptional activation of its target genes. Haa1 is a multiply phosphorylated protein [18,19]. Haa1 phosphorylation is partially mediated by Hrr25, one of the four isoforms of casein kinase protein in *S. cerevisiae* [19]. In *hrr25* mutant cells, Haa1 is hypophosphorylated, which correlates with its increased nuclear localization and higher levels of expression of its target genes. Nucleocytoplasmic shuttling is mediated by specific importins and exportins (reviewed in [20]) and Msn5 is required for the nuclear export of Haa1 [18]. 

Protein kinase A is an important regulator of cell growth, metabolism, and stress responses (reviewed in [21]). In the presence of high-quality nutrients, PKA increases the expression of ribosome biogenesis and positively regulates cell growth and proliferation [22,23,24,25]. When nutrients are limited, reduced PKA activity allows cells to adapt to various stress conditions. PKA has been implicated in yeast cell death and apoptosis in the presence of lethal concentrations of acetic acid [10,26,27]. There is a report suggesting that PKA is involved in Haa1-mediated acetic acid stress response, but its role has not been directly examined [28,29]. The protein kinase, Yak1, has been proposed to work downstream or in a parallel pathway of PKA [28,30,31], and Malcher et al. [28] reported that Haa1 and Yak1 regulate the expression of a few common target genes. Here, we analyzed the effect of mutations that alter PKA activity in *S. cerevisiae* on the expression of Haa1-regulated genes and cellular fitness in response to acetic acid treatment. We uncovered a negative regulatory role of PKA in the acetic acid stress response in yeast. 

## 2. Materials and Methods

### 2.1. Strains, Growth Media, and Growth Conditions

Yeast strains were grown at 30 °C in a YPD medium (1% bacto yeast extract (Fisher Scientific, Waltham, MA, USA), 2% bacto peptone (Fisher Scientific), 2% glucose (Fisher Scientific)), a YNBcasD medium (0.67% yeast nitrogen base (Fisher Scientific), 1% casamino acids (Fisher Scientific), 2% dextrose), synthetic dextrose minimal media (SD) (0.67% yeast nitrogen base without amino acids, 2% glucose), and an MM4 medium (0.17% yeast nitrogen base without amino acids and ammonium sulfate (Fisher Scientific), 0.265% ammonium sulfate (Fisher Scientific), 2% glucose, adjusted to pH 4) with or without 60 mM of acetic acid (Fisher Scientific). The SD and MM4 media were supplemented with uracil, leucine, histidine, methionine, and lysine (Sigma-Aldrich, St. Louis, MO, USA) at standard concentrations to cover auxotrophic requirements when required [32]. Agar (USBiological, Salem, MA, USA) was added at a final concentration of 2% for the solid medium. The yeast strains used in this study are listed in Table 1. Yeast strains carrying a hyperactive *RAS2^A18V19^* allele were obtained by transformation with StuI-linearized pRS303-RAS2A18V19 (pZL3326). Transformants were selected on the SD medium without histidine. Yeast strains with *TPO2*, *YRO2*, *YGP1–lacZ* reporter genes integrated in the nuclear genome were obtained by transformation with respective *lacZ* reporter gene plasmids linearized by digesting with a restriction enzyme at a unique site. Transformants were selected on the SD medium without leucine (integrated *TPO2–lacZ* reporter gene) or on a YPD medium supplemented with 300 mg/L geneticin (integrated *YRO2–lacZ* or *YGP1–lacZ* reporter gene). Other deletion mutant strains were constructed by transforming yeast with respective gene knockout cassettes, dissecting sporulated heterozygous diploid strains, and/or crossing two mutant strains to obtain diploids for sporulation and dissection. Gene deletion mutations were confirmed by PCR genotyping. microorganisms-12-01452-t001_Table 1Table 1Yeast strains used in this study.StrainGenotypeSourceApplicationBY4741*
MATa his3
*
Δ*1 leu2*Δ*0 met17*Δ*0 ura3*Δ*0*
Lab. stockFigure 1A–D, Figure 2, Figure 3, Figure 4, Figure 5 and Figure 6ZLY4370BY4741 *ras1*Δ*::kanMX4*This studyFigure 1A–DZLY4376BY4741 *ras2*Δ*::kanMX4-2D*This studyFigure 1A–D, Figure 2 and Figure 3ZLY4373BY4741 *pde1*Δ*::kanMX4*This studyFigure 1A–DZLY4368BY4741 *pde2*Δ*::kanMX4*This studyFigure 1A–D and Figure 3ZLY4162BY4741 *YRO2–lacZ::kanMX4*This studyFigure 1E,FZLY4153BY4741 *YGP1–lacZ::kanMX4*This studyFigure 1E,FZLY4238BY4741 *TPO2–lacZ::LEU2*This studyFigure 1E,FZLY4322BY4741 *YRO2–lacZ::kanMX4 RAS2::RAS2Ala18Val19::HIS3*This studyFigure 1E,FZLY4319BY4741 *YGP1–lacZ::kanMX4 RAS2::RAS2Ala18Val19::HIS3*This studyFigure 1E,F and Figure 3ZLY4323BY4741 *TPO2–lacZ::LEU2 RAS2::RAS2Ala18Val19::HIS3*This studyFigure 1E,FZLY4043BY4741 *haa1*Δ*::kanMX4*SGDP *Figure 2, Figure 3, Figure 5 and Figure 6ZLY4470BY4741 *haa1::HIS3 ras2::kanMX4*This studyFigure 2ZLY4419 ***
MATa his3
*
Δ*1 leu2*Δ*0 met17*Δ*0 ura3*Δ*0*
This studyFigure 3CZLY4424 ***
MATa his3
*
Δ*1 leu2*Δ*0 met17*Δ*0 ura3*Δ*0*
This studyFigure 3CZLY4411BY4741 *ras2*Δ*::kanMX4-1D*This studyFigure 3CZLY4445BY4741 *tpk1::kanMX4*This studyFigure 4ZLY4438BY4741 *tpk2::kanMX4*This studyFigure 4ZLY4441BY4741 *tpk3::kanMX4*This studyFigure 4ZLY4516*
MATa ura3
*
Δ *leu2*Δ *his3*Δ *tpk1::kanMX4 tpk2::kanMX4*
This studyFigure 4 and Figure 5ZLY4518*
MATα ura3
*
Δ *leu2*Δ *his3*Δ *met17*Δ *lys2*Δ *tpk1::kanMX4 tpk2::kanMX4*
This studyFigure 4ZLY4520*
MATa ura3
*
Δ *leu2*Δ *his3*Δ *lys2*Δ *tpk1::kanMX4 tpk3::kanMX4*
This studyFigure 4, Figure 5 and Figure 6ZLY4521*
MATα ura3
*
Δ *leu2*Δ *his3*Δ *met17*Δ *tpk1::kanMX4 tpk3::kanMX4*
This studyFigure 4ZLY4522*
MATα ura3
*
Δ *leu2*Δ *his3*Δ *met17*Δ *tpk2::kanMX4 tpk3::kanMX4*
This studyFigure 4 and Figure 5ZLY4524BY4741 *tpk2::kanMX4 tpk3::kanMX4*This studyFigure 4ZDY166BY4741 *haa1::HIS3*This studyFigure 5FZLY5264*
MATa ura3
*
Δ *leu2*Δ *his3*Δ *lys2*Δ *tpk1::kanMX4 tpk3::kanMX4 haa1::LEU2*
This studyFigure 5 and Figure 6*, *Saccharomyces* genome deletion project [33]; **, this strain was obtained from tetrad analysis and had the same genotype as BY4741.

### 2.2. Plasmid Constructs

The plasmids used in this study are listed in Table 2. To generate the *YGP1–lacZ* reporter gene, two primers, 5′-GTCAGAATTCTTGGTAGACATGGTGGTGC-3′ and 5′-GTCAGTCGACCAGATAAAACAACTTGGAACTTCAT-3′, were used to amplify the DNA sequence from position −1980 to 25 in the promoter region of *YGP1* using BY4741 genomic DNA as the template. The resulting PCR product was digested with EcoRI and SalI, and fused in frame to the *E. coli LacZ* gene in the centromeric plasmid pWEJ [34] to form plasmid pZL3170. To generate the *TPO3–lacZ* reporter gene in plasmid pZL3161, two primers, 5′-GTCAGGATCCTTGTCGAAATCGTCCAA-3′ and 5′-GTCAAAGCTTTGGATTCCTGTCTGTTCATTTC-3′, were used to amplify the DNA sequence from position −1976 to 19 in the promoter region of *TPO3,* which was cloned into the BamHI and HindIII sites of pWEJ. Similarly, two primers, 5′-GTCAGGATCCAATGGATACTCTTCGTATCG-3′ and 5′-GTCAAAGCTTGGAGCTTAGCGTTAGACAACAT-3′, were used to amplify the DNA sequence from position −980 to 22 in the promoter region of *SPI1* for the construction of the *SPI1–lacZ* reporter gene plasmid pZL3155. pRS303-RAS2^A18V19^ (pZL3326) was constructed by cloning an EcoRI and BamHI-digested fragment encoding the hyperactive *RAS2^A18V19^* allele from YCp50–RAS2^A18V19^ [35] into the corresponding cloning sites of pRS303. To generate YIp356–kanMX4, an integrative *lacZ* reporter gene plasmid carrying the *kanMX4* selection marker, a portion of the *URA3* open reading frame in YIp356 [36] was removed by digesting with NcoI and StuI and replaced by a Klenow-fragment-filled *kanMX4* cassette digested with EcoRI and BglII. Integrative *YRO2* and *YGP1–lacZ* reporter plasmids (pZL3203 and pZL3201) were constructed by cloning their respective promoters into YIp356–kanMX4. The integrative *TPO2–lacZ* reporter plasmid pZL3307 was generated by cloning the *TPO2* promoter sequence into YIp366 [36]. The restriction enzymes were purchased from New England Biolabs (Ipswich, MA, USA).microorganisms-12-01452-t002_Table 2Table 2Plasmids used in this study.PlasmidDescriptionReferenceApplicationpZL3164pRS416-YRO2–lacZ, expressing *YRO2–lacZ* reporter gene with 1929-bp *YRO2* promoter sequence fused to *lacZ* coding sequence.[19]Figure 1A–D, Figure 2, Figure 4 and Figure 5pZL3170pRS416-YGP1–lacZ, expressing *YGP1–lacZ* reporter gene with 1980 bp *YGP1* promoter sequence fused to *lacZ* coding sequence.This studyFigure 1A–D, Figure 2, Figure 4 and Figure 5YCp50-RAS2^A18V19^A hyperactive RAS2 mutant allele, RAS2A18V19, cloned in centromeric plasmid with a *URA3* selection marker.[35]
pZL3326pRS303-RAS2^A18V19^, encoding the dominant active *RAS2^A18V19^* allele on an integrative plasmid carrying a *HIS3* selection marker.This studyFigure 1E,F and Figure 3YIp356-kanMX4An integrative plasmid carrying a *lacZ* reporter gene and the kanMX4 selection marker.This study
pZL3203YIp356-kanMX4-YRO2, encoding an *YRO2–lacZ* reporter gene for integration into the genome.This studyFigure 1E,FpZL3201YIp356-kanMX4-YGP1, encoding an *YGP1–lacZ* reporter gene for integration into the genome.This studyFigure 1E,FpZL3307YIp366-TPO2, encoding an *TPO2–lacZ* reporter gene for integration into the genome.This studyFigure 1E,FpZL3158pRS416-TPO2–lacZ, expressing *TPO2–lacZ* reporter gene with 1926 bp TPO2 promoter sequence fused to the *lacZ* coding sequence.[19]Figure 2 and Figure 5pZL3161pRS416-TPO3–lacZ, expressing *TPO3–lacZ* reporter gene with 1976 bp *TPO3* promoter sequence fused to the *lacZ* coding sequence.This studyFigure 5pZL3155pRS416-SPI1–lacZ, expressing *SPI1–lacZ* reporter gene with 980 bp *SPI1* promoter sequence fused to the *lacZ* coding sequence.This studyFigure 5

### 2.3. Yeast Transformation and β-Galactosidase Activity Assays 

For transformation, yeast cells were freshly grown in YPD liquid medium and transformed using a high-efficiency method [37]. The YNBcasD medium, the SD medium supplemented with appropriate amino acids and uracil, and the YPD medium supplemented with geneticin were used to select yeast transformants based on the *URA3*, *LEU2*, and *kanMX4* selection markers, respectively. For the β-galactosidase activity assays, the yeast strains were grown in the MM4 medium with or without 60 mM acetic acid at 30 °C for at least 6 generations to reach an OD_600_ of 0.5–0.8 before collection. The cells were collected by centrifugation, and the β-galactosidase activity assays were conducted as described [32]. During the pilot stage of the reporter gene analysis, a minimum of two independently obtained mutant strains were utilized in the analysis of the effect of a specific mutation on the expression of *lacZ* reporter genes and usually found to yield similar results. Two to six independent cultures were grown, and assays were carried out in duplicate for each sample. The data are presented as the mean ± standard deviation. The means of the β-galactosidase activity assay results were compared using the *t*-test. The “*” in figures indicates a significant difference in the means of two groups of data (*p* < 0.05). The two ends of a horizontal line under an “*” in the data figures mark the two strains under comparison. 

### 2.4. Generation of Growth Curves

The wild-type and isogenic mutant strains were grown on the YPD solid medium for 2 to 3 days at 30 °C. The cells from the plate were then grown in the liquid MM4 medium overnight to an OD_600_ of 1~1.6 to obtain pre-cultures, which were diluted into the MM4 medium with and without 60 mM acetic acid with a starting OD_600_ of ~0.05. The OD_600_ values of the cell cultures were determined over time until they reached the stationary phase. 

### 2.5. Serial Dilution of Cells for Growth Analysis

The wild-type and isogenic mutant strains were freshly grown on the YPD solid medium at 30 °C for 2–3 days. The cells were picked from a plate into sterile water and diluted to the same starting OD_600_ of 0.1. Five-fold serial dilutions were made using sterile 96-well plates and 8-channel pipettes. As described in the figure legend, 5 μL aliquots were spotted on solid medium. The cells were grown for 2 to 4 days at 30 °C before pictures were taken for cell growth analysis. 

### 2.6. Differential Interference Contrast Microscopy of Yeast Cultures

The cells were grown in the MM4 medium without or with 60 mM acetic acid for a minimum of six generations to reach OD_600_ ≥ 0.6. Before coverslips were applied, 15 μL cell cultures were pipetted onto the slide for 3 min. Differential interference contrast (DIC) images of live cells were immediately captured using a Nikon Eclipse E800 microscope equipped with a Nikon Plan Fluor 40× DIC M objective lens (Nikon Inc., Melville, NY, USA). Images were acquired with a Photometrics Coolsnap fx charge-coupled device (CCD) camera and Metamorph imaging software (version 7.7) (Molecular Devices, Sunnyvale, CA, USA) and processed using ImageJ (version 1.53k) (National Institutes of Health, Bethesda, MD, USA).

## 3. Results

### 3.1. ras2*Δ* and pde2*Δ* Increase and Decrease the Expression of YRO2 and YGP1–lacZ Reporter Gene, Respectively

To determine the potential role of PKA in the regulation of Haa1-dependent gene expression, we analyzed the expression of *lacZ* reporter genes driven by the promoters of two well-established target genes of Haa1, *YRO2* and *YGP1* [3,38], in mutants with altered PKA activities. PKA is activated by the second messenger cyclic AMP, which is produced by adenylate cyclase (reviewed in [21]). Adenylate cyclase is activated by two Ras proteins, Ras1 and Ras2 [39]. The downregulation of PKA is achieved via two cyclic AMP phosphodiesterases, Pde1 and Pde2, which convert cyclic AMP to AMP [40]. We generated *pde1*Δ, *pde2*Δ, *ras1*Δ, and *ras2*Δ mutants by dissecting sporulated heterozygous diploid mutant strains and transformed them with centromeric plasmids encoding a *YRO2–lacZ* or *YGP1–lacZ* reporter gene. The transformants were grown in MM4 medium without or with 60 mM acetic acid for a minimum of six generations to reach the logarithmic phase, and the cells were collected for β-galactosidase activity assays. In the cells grown in the absence of acetic acid supplementation, *pde2*Δ reduced the expression of *YRO2–lacZ* by 4.3-fold, while *ras2*Δ increased its expression by 2.8-fold (Figure 1A). In the cells grown in the presence of 60 mM acetic acid, the overall effect of *pde2*Δ and *ras2*Δ on the expression of *YRO2–lacZ* reporter gene was still present, albeit at reduced levels: *pde2*Δ reduced *YRO2–lacZ* expression by 27%, while *ras2*Δ increased its expression by 2.1-fold (Figure 1B). Under both growth conditions, *pde1*Δ and *ras1*Δ did not significantly affect the expression of the *YRO2–lacZ* reporter gene. Similar results were obtained for the *YGP1–lacZ* reporter gene (Figure 1C,D). In the MM4 medium without acetic acid supplementation, *pde2*Δ reduced *YGP1–lacZ* expression by 8-fold, while *ras2*Δ increased its expression by 4.2-fold. In the MM4 medium supplemented with 60 mM acetic acid, *pde2*Δ reduced *YGP1–lacZ* expression by 20%, and *ras2*Δ increased its expression by 2.2-fold. As was observed for the *YRO2–lacZ* reporter gene, *pde1*Δ and *ras1*Δ did not result in significant changes in the expression of the *YGP1–lacZ* reporter gene. Our findings are consistent with the notion that PKA negatively regulates the expression of Haa1-regulated genes.
Figure 1The opposite effects of *pde2*Δ and *RAS2^A18V19^* versus *ras2*Δ on the expression of *YRO2–lacZ*, *YGP1–lacZ*, and/or *TPO2–lacZ* reporter genes. (**A**–**D**) *pde2*Δ and *ras2*Δ have opposite effects on the expression of *YRO2–lacZ* and *YGP1–lacZ* reporter genes. Wild type (BY4741) and isogenic *pde1*Δ (ZLY4373), *pde2*Δ (ZLY4368), *ras1*Δ (ZLY4370), *and ras2*Δ (ZLY4376) mutant strains carrying a centromeric plasmid encoding a *YRO2–lacZ* (pZL3164, panels (**A**,**B**)) or *YGP1–lacZ* reporter gene (pZL3170, panels (**C**,**D**)) were grown in the MM4 medium without acetic acid (panels (**A**,**C**)) and with 60 mM acetic acid (panels (**B**,**D**)) to mid-logarithmic phase. β-galactosidase activity assays were conducted as described in Materials and Methods. The data are presented as the mean ± standard deviation. The means of the results were compared by *t*-test. “*” indicates a significant difference (*p* < 0.05) in the means of two groups of data indicated by the beginning and end of a horizontal line throughout this manuscript. (**E**,**F**) A dominant, hyperactive allele of *RAS2*, *RAS2^A18V19^*, decreases the expression of *YRO2–*, *YGP1–*, and *TPO2–lacZ* reporter genes. The yeast strains carrying an integrated reporter gene as indicated (*YRO2–lacZ*, ZLY4162; *YGP1–lacZ*, ZLY4153; *TPO2–lacZ*, ZLY4238) were transformed with an integrative plasmid encoding *RAS2^A18V19^*. The resultant transformants and non-transformed control strains were grown in the MM4 medium without (panel (**E**)) and with (panel (**F**)) 60 mM acetic acid. β-galactosidase activity assays were conducted.
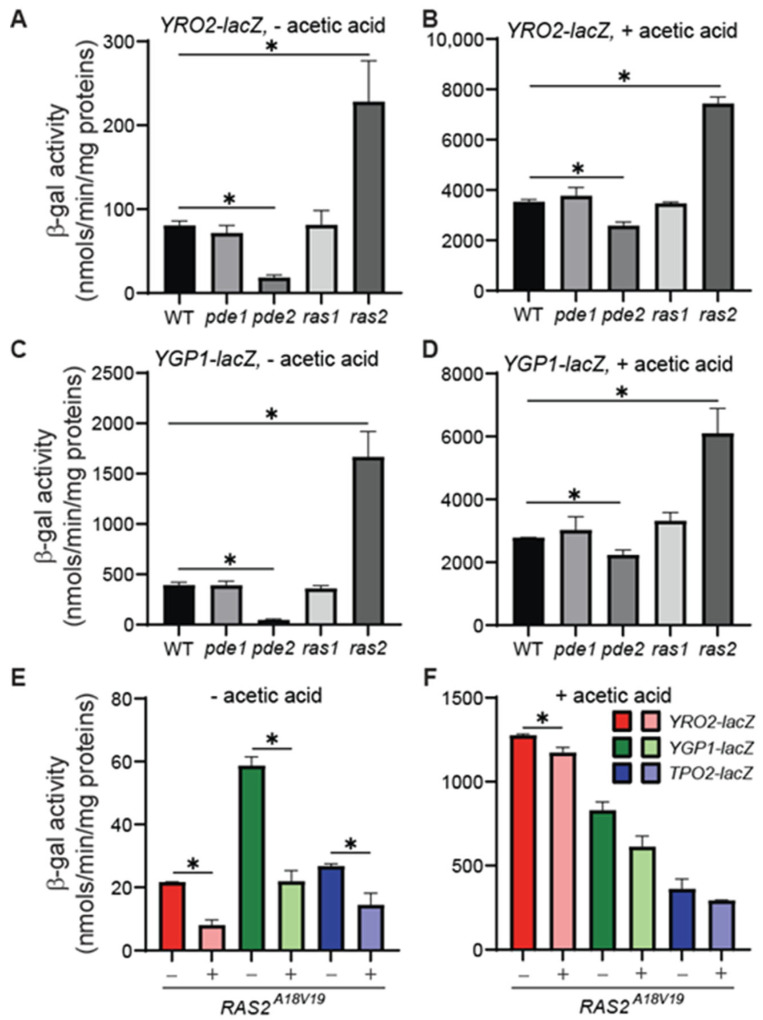


### 3.2. A Hyperactive Allele of RAS2, RAS2^A18V19^, Reduces the Expression of YRO2, YGP1, and TPO2–lacZ Reporter Genes

Ras2 and Pde2 are a positive and a negative regulator of PKA, respectively. To further test the role of PKA in the regulation of the expression of Haa1-dependent genes, we examined the effect of a hyperactive allele of *RAS2*, *RAS2^A18V19^* [41,42] expressed from the low-copy centromeric plasmid YCp50. The glycine 19 to valine mutation in *RAS2* leads to constitutive activation of PKA signaling. The *lacZ* reporter genes described in Figure 1A–D are encoded on the centromeric plasmid pRS416, which has the same *URA3* selection marker as YCp50. Thus, we generated *YRO2–lacZ* and *YGP1–lacZ* reporters on integrative plasmids using the *kanMX4* selection marker and integrated them into the nuclear genome. We added another Haa1-regulated gene, *TPO2*, and constructed an integrative *TPO2–lacZ* reporter gene carrying the *LEU2* selection marker. The yeast strains carrying these integrated reporter genes showed a more than 12-fold increase in β-galactosidase activities in response to the treatment with 60 mM acetic acid (compare Figure 1E and Figure 1F), indicating that these three integrated *lacZ* reporter genes were valid readouts of the acetic acid stress response. Our initial analysis showed that the expression of *RAS2^A18V19^* on YCp50 significantly reduced the expression of all three *lacZ* reporter genes in cells grown in the absence of acetic acid treatment. However, we noticed it took 2–3 times longer for the cells carrying the plasmid YCp50–RAS2^A18V19^, in comparison with an empty vector, to undergo six cell divisions in the MM4 medium supplemented with 60 mM acetic acid to reach the same OD_600_ reading of ~0.6. The growth defects of yeast strains grown in the presence of the 60 mM acetic acid due to the expression of *RAS2^A18V19^* will be described later in Figure 3. We were concerned that the growth defect might affect the yeast cell’s ability to maintain the YCp50–RAS2^A18V19^ plasmid, which could complicate the interpretation of reporter gene activity data. Therefore, we generated an integrative plasmid encoding *RAS2^A18V19^* and integrated it into the *RAS2* genomic locus. The β-galactosidase activity assay showed that *RAS2^A18V19^* significantly reduced the expression of *YRO2–*, *YGP1*–, and *TPO2–lacZ* reporter genes in cells grown in MM4 medium without acetic acid treatment (Figure 1E). In cells grown in the MM4 medium supplemented with 60 mM acetic acid, *RAS2^A18V19^* marginally reduced the expression of these three reporter genes. Together, these data suggest that the activation of PKA due to *pde2*Δ or *RAS2^A18V19^* reduces the expression of Haa1-regulated genes. Conversely, reduced PKA activity due to a *ras2*Δ mutation increases the expression of Haa1-regulated genes. 

### 3.3. Increased Expression of YRO2–, YGP1–, and TPO2–lacZ Reporter Genes Due to ras2*Δ* Requires Haa1

Our data from Figure 1 suggest that PKA negatively regulates the expression of *YRO2*, *YGP1*, and *TPO2*, which are known target genes of Haa1. These target genes may require transcription factors other than Haa1. Thus, PKA may negatively regulate Haa1 or other transcription factors. To differentiate these two possibilities, we determined whether the increased expression of *YRO2–*, *YGP1–*, and *TPO2–lacZ* reporter genes due to *ras2*Δ required Haa1. Consistent with published findings, *haa1*Δ reduced the expression of these three reporter genes under both basal and acetic acid-inducing conditions (Figure 2). The increased expression of *YRO2–lacZ* and *TPO2–lacZ* due to *ras2*Δ was largely reversed by *haa1*Δ under both basal and acetic acid-inducing conditions. In contrast, the *ras2*Δ-induced expression of the *YGP1–lacZ* reporter gene was only partially reversed by *haa1*Δ. *YGP1–lacZ* has higher basal activity (compare Figure 2C with Figure 2A,E), suggesting that other transcription factors also contribute to *YGP1* expression. The (partial) reversal of *ras2*Δ-induced expression of *YRO2*, *TPO2*, and *YGP1* by *haa1*Δ suggest that *ras2*Δ *haa1*Δ mutant cells may be sensitive to acetic acid treatment. Figure 2G shows that this is indeed the case. *ras2*Δ led to slight growth defects on the solid MM4 medium. Surprisingly, *ras2*Δ mutant cells did not exhibit increased resistance to acetic acid despite having an increased expression of *YRO2*, *YGP1*, and *TPO2*. This paradox will be explained in the section below. Together, our data suggest that *ras2*Δ-induced expression of *YRO2*, *YGP1*, and *TPO2* is (partially) dependent on Haa1. Figure 2Haa1 is required or partially required for increased expression of *YRO2–lacZ*, *YGP1–lacZ*, and *TPO2–lacZ* reporter genes due to *ras2*Δ. Wild type (BY4741) and isogenic *haa1*Δ (ZLY4043), *ras2*Δ (ZLY4376), and *ras2*Δ *haa1*Δ (ZLY4470) mutant strains carrying a centromeric plasmid encoding a *YRO2–lacZ* (pZL3164, panels (**A**,**B**)), *YGP1–lacZ* (pZL3170, panels (**C**,**D**)), or *TPO2–lacZ* (pZL3158, panels (**E**,**F**)) reporter gene were grown in the MM4 medium without (panels (**A**,**C**,**E**)) and with 60 mM acetic acid (panels (**B**,**D**,**F**)). β-galactosidase activity assays were conducted. The data are presented as the mean ± standard deviation. (**G**) *ras2*Δ *haa1*Δ double mutant cells are sensitive to acetic acid stress. Wild-type and isogenic mutant strains described for panels (**A**–**F**) were serially diluted and spotted on MM4 medium without and with 60 mM acetic acid. “*” indicates a significant difference (*p* < 0.05) in the means of two groups of data indicated by the beginning and end of a horizontal line.
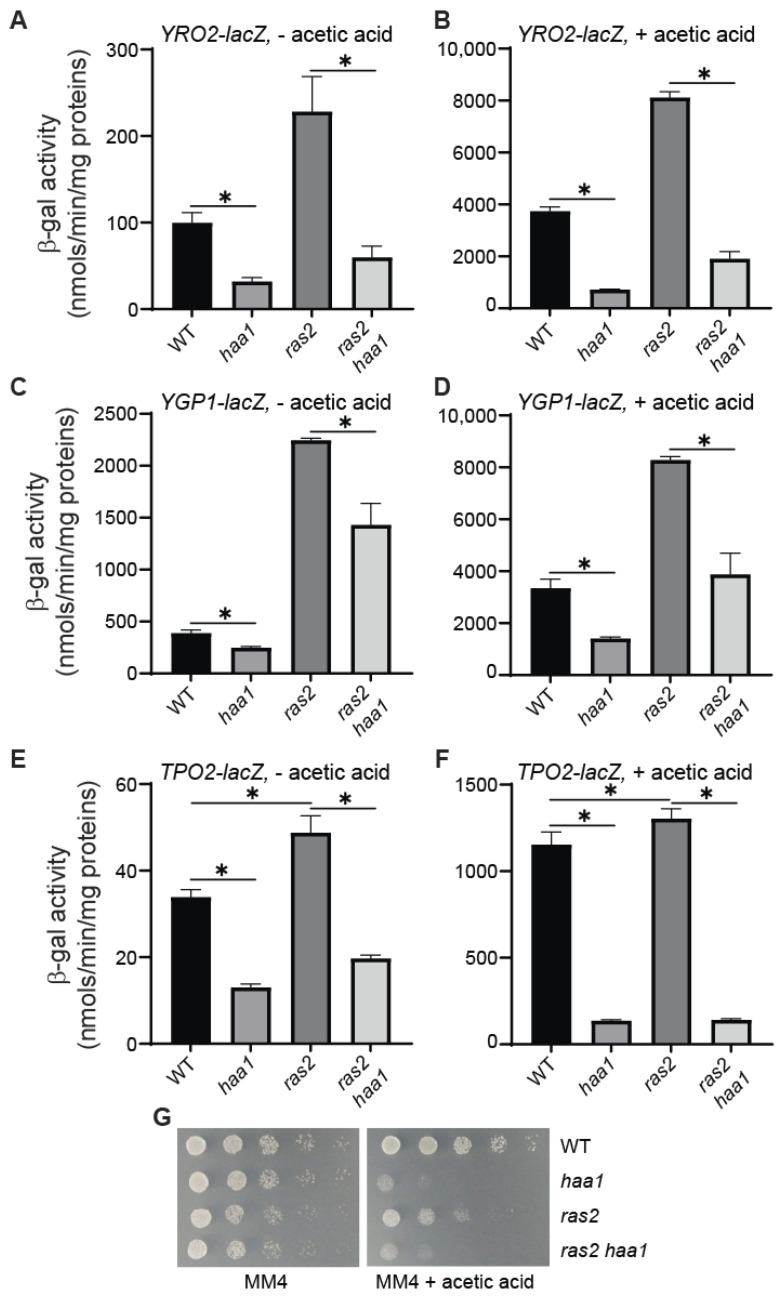


### 3.4. pde2*Δ* and RAS2^A18V19^ Increase Acetic Acid Sensitivity While ras2*Δ* Improves Cellular Fitness in Response to Acetic Acid Stress

Haa1 is required for cellular adaptation to acetic acid stress. Our data thus far suggest that PKA negatively regulates Haa1. Altered Haa1 activity in mutants that affect PKA signaling may result in changes in cellular fitness in the presence of acetic acid. To test this possibility, we generated and compared the growth curves of wild-type, *haa1*Δ, *RAS2^A18V19^*, *pde2*Δ, and *ras2*Δ mutant strains grown in the MM4 medium without and with 60 mM acetic acid. In the absence of acetic acid treatment, *haa1*Δ, *pde2*Δ, and *RAS2^A18V19^* mutant cells exhibited similar growth to the wild-type strain (Figure 3A). In contrast, in response to acetic acid treatment, *pde2*Δ and *RAS2^A18V19^* mutant cells had similar growth defects to *haa1*Δ mutant cells (Figure 3B). In the absence of acetic acid treatment, *ras2*Δ led to slight growth defects (Figure 3A), which is consistent with the growth defects of *ras2*Δ mutants on the MM4 solid medium lacking acetic acid (Figure 2G). In the presence of acetic acid treatment, *ras2*Δ abolished the lag phase in the initial stage of acetic acid treatment, and *ras2*Δ mutant cells outperformed wild type (Figure 3B). We further compared the growth curves of four wild-type strain cultures (three independently obtained wild-type strains) and three *ras2*Δ mutant strain cultures (two independently obtained *ras2* mutant strains) in the MM4 medium supplemented with 60 mM acetic acid and found that the phenotype was reproducible (Figure 3C). 

Sensitivity to acetic acid treatment due to *haa1*Δ is one of the most prominent phenotypes in the acetic acid stress response pathway in *S. cerevisiae*. Reduced cellular fitness in *pde2*Δ and *RAS2^A18V19^* mutant cells in response to acetic acid treatment is consistent with our model that PKA is a negative regulator of Haa1. Improved cellular fitness of *ras2*Δ mutant cells in response to acetic acid stress further lends support to this model. When yeast cells encounter weak organic acids at inhibitory concentrations, cells arrest growth initially and may resume exponential growth after an extended lag phase (reviewed in [1]). When the adapted cells are transferred to a fresh medium with the same level of weak acids, no growth delay is detected. *ras2*Δ cells in Figure 3 resemble wild-type cells that have adapted to acetic acid stress. This is consistent with our notion that PKA is a negative regulator of acetic acid stress in yeast.

If *ras2*Δ abolishes the lag phase as seen in Figure 3B,C, should not the *ras2*Δ mutant cells show a relative growth advantage over the wild-type on the MM4 solid medium supplemented with 60 mM acetic acid in Figure 2G? This could be reconciled by the difference in yeast cell growth on the solid medium versus the liquid medium. In the serial dilution analysis of cell growth presented in Figure 2G, cells were picked from the plate after 2–3 days on the YPD medium and transferred into water for determining OD_600_. Almost all cells from the YPD plate were unbudded at this stage. It is possible that both the wild-type and *ras2*Δ mutant cells have low PKA activity at this stage of cell growth. Their transfer to water for serial dilutions may further reduce the difference, if any, in PKA activity. This would lead to the observed result in Figure 2G: the *ras2*Δ cells do not lead to a growth advantage over the wild type on the MM4 medium supplemented with acetic acid. In Figure 3, the precultures were grown in the MM4 liquid medium overnight to the mid–late logarithmic phase before they were diluted into the MM4 medium without and with acetic acid to initiate the experiment for the generation of growth curves. It is expected that the PKA pathway activities are different between the wild-type and *ras2*Δ mutant cells in the logarithmic phase of cell growth in the dextrose medium. Figure 3Mutations in *PDE2* and *HAA1*, and expression of the hyperactive *RAS2^A18V19^* allele reduce cellular fitness in response to acetic acid stress, while *ras2*Δ increases it. (**A**,**B**) The growth curves of wild-type and isogenic mutant strains as indicated were generated from cells grown in MM4 medium without acetic acid (panel (**A**)) and with 60 mM acetic acid (panel (**B**)). The growth curves are representative of data from two independent cultures. WT, BY4741; *haa1*Δ, ZLY4043; *RAS2^A18V19^*, ZLY4319; *pde2*Δ, ZLY4368; *ras2*Δ, ZLY4376. (**C**) The growth curves of four cultures of wild type and three cultures of *haa1*Δ mutant strains in MM4 medium supplemented with 60 mM acetic acid. WT #1 and #2, BY4741; WT #3, ZLY4419; WT #4, ZLY4424; *ras2* #1 and #2, ZLY4376; *ras2* #3, ZLY4411.
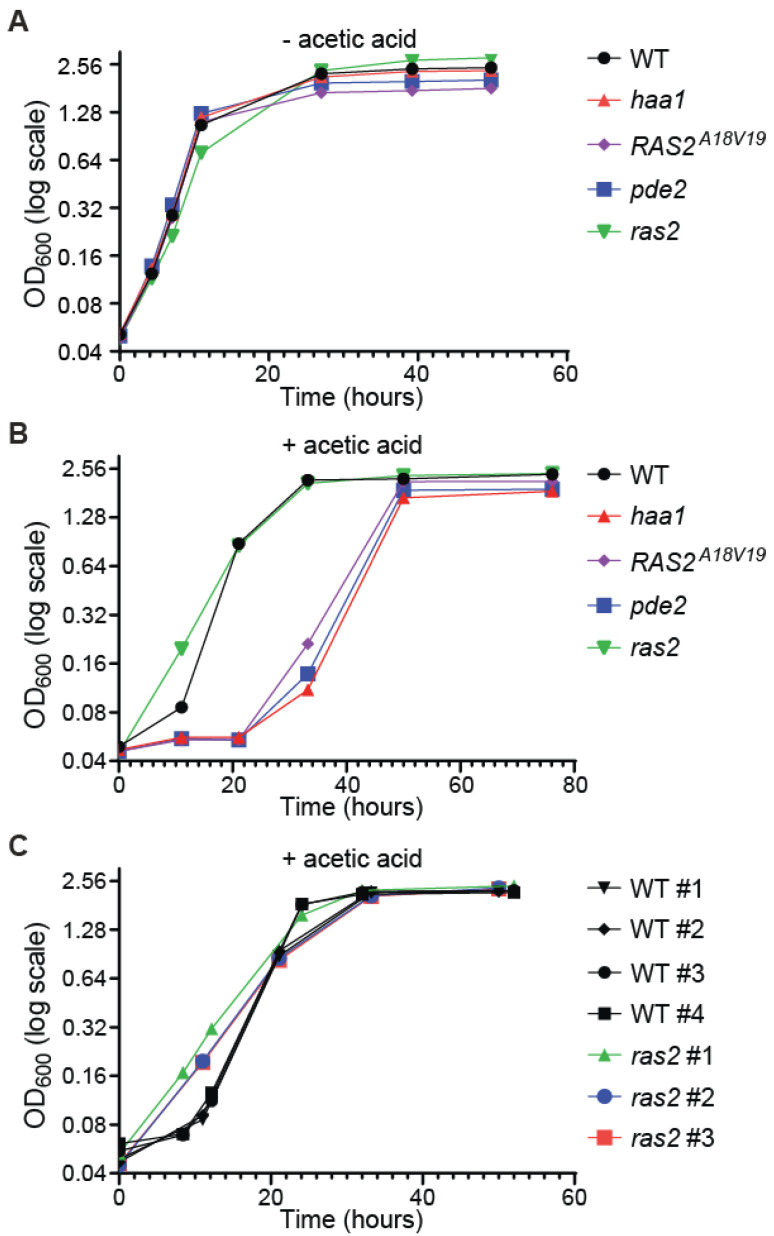


### 3.5. Mutations in TPK1 (TPK3) and TPK2 Have Opposite Effects on YRO2 and YGP1 Expression

PKA has three catalytic subunits, Tpk1, Tpk2, and Tpk3 [43]. Although they are redundant for cell viability, these yeast Tpk isoforms are known to have nonredundant, pathway-specific roles (reviewed in [44]). To determine which catalytic subunit is involved in the regulation of acetic acid stress responsive genes, we analyzed the expression of *YRO2–* and *YGP1–lacZ* reporter genes in strains carrying a single deletion of *TPK1*, *TPK2*, or *TPK3*. In cells grown in the absence of acetic acid treatment, *tpk1*Δ and *tpk3*Δ single deletion strains led to an increased expression of both *YRO2–* and *YGP1–lacZ* reporter genes, while *tpk2*Δ reduced their expression (Figure 4A,C)*.* Between *tpk1*Δ and *tpk3*Δ, *tpk3*Δ had a stronger effect than *tpk1*Δ, suggesting that Tpk3 is a stronger inhibitor of *YRO2* and *YGP1* expression than Tpk1. In the cells grown in the presence of 60 mM acetic acid, the inducing effects due to *tpk1*Δ and *tpk3*Δ single mutations were largely maintained, and the effect of *tpk2*Δ on the expression of *YRO2–* and *YGP1–lacZ* reporter genes was abolished or reversed (Figure 4B,D). Figure 4Mutations in *TPK1* (*TPK3*) and *TPK2* have opposite effects on the expression of *YRO2* and *YGP1*. (**A**–**D**) The effect of *tpk1*Δ, *tpk2*Δ, and *tpk3*Δ single mutations on the expression of *YRO2–lacZ* and *YGP1–lacZ* reporter genes. Wild-type (BY4741) and isogenic *tpk1*Δ (ZLY4445), *tpk2*Δ (ZLY4438), and *tpk3*Δ (ZLY4441) mutant strains carrying a centromeric plasmid encoding a *YRO2–lacZ* (pZL3164, panels (**A**,**B**)) or *YGP1–lacZ* (pZL3170, panels (**C**,**D**)) reporter gene were grown in MM4 medium without acetic acid (panels (**A**,**C**)) and with 60 mM acetic acid (panels (**B**,**D**)) to mid-logarithmic phase. β-galactosidase activity assays were conducted. The data are presented as the mean ± standard deviation. (**E**–**H**) The effect of *tpk1/2*Δ, *tpk1/3*Δ, and *tpk2/3*Δ double mutations on the expression of *YRO2–lacZ* and *YGP1–lacZ* reporter genes. Wild type (BY4741) and isogenic *tpk1/2*Δ (ZLY4516 and ZLY4518), *tpk1/3*Δ (ZLY4520 and ZLY4521), and *tpk2/3*Δ (ZLY4522 and ZLY4524) were analyzed for the expression of *YRO2–lacZ* and *YGP1–lacZ* reporter genes as described for panels (**A**–**D**). The data are presented as the mean ± standard deviation. “*” indicates a significant difference (*p* < 0.05) in the means of two groups of data indicated by the beginning and end of a horizontal line.
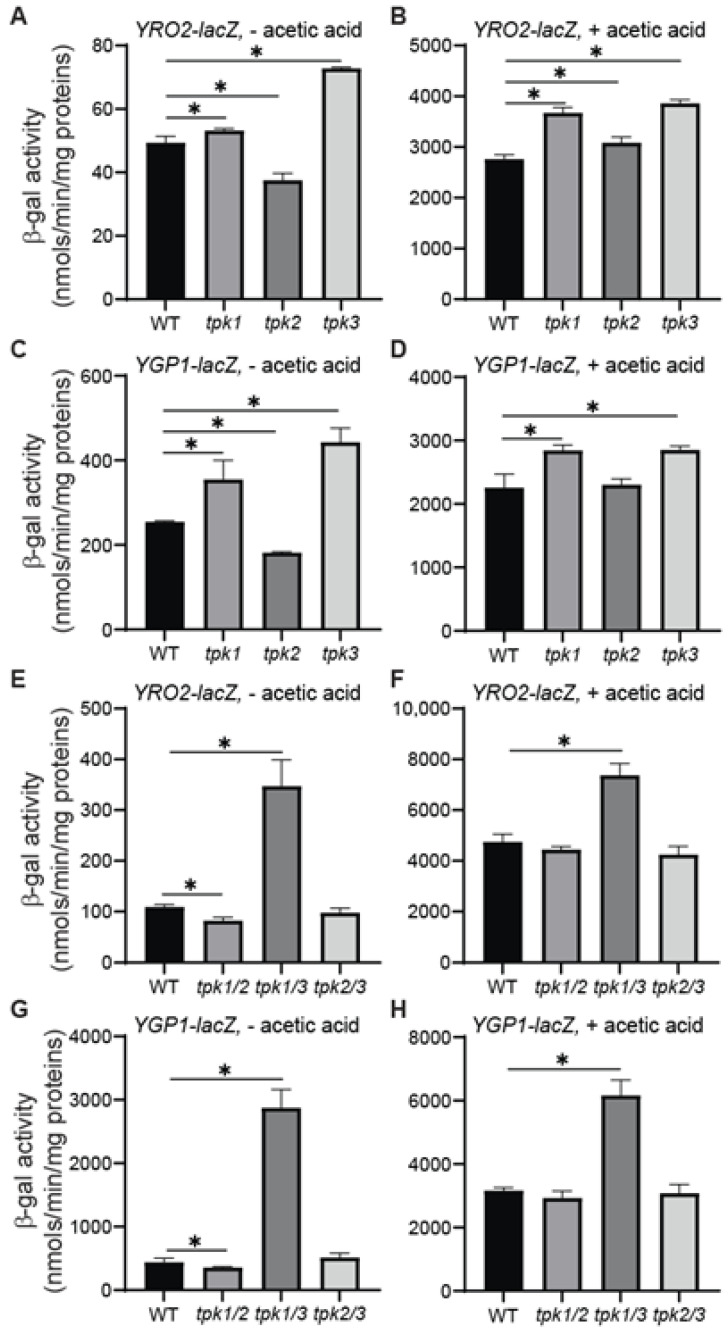


We next examined the effect of *tpk1*Δ *tpk2*Δ*,* *tpk1*Δ *tpk3*Δ*,* and *tpk2*Δ *tpk3*Δ double deletions on the expression of *YRO2* and *YGP1* using β-galactosidase activity assays. Consistent with a mild induction in the expression of *YRO2–* and *YGP1–lacZ* reporter genes in the *tpk1*Δ and *tpk3*Δ single deletion strains, a *tpk1*Δ *tpk3*Δ double deletion further increased their expression under both basal and acetic acid stress conditions (compare Figure 4E,F to Figure 4A,B and Figure 4G,H to Figure 4C,D). Additionally, *tpk2*Δ reversed the increased expression of *YRO2–* and *YGP1–lacZ* reporter genes in *tpk3*Δ mutant cells to the wild-type levels under both growth conditions. *tpk2*Δ also reversed the increased expression of *YRO2–* and *YGP1–lacZ* reporters due to *tpk1*Δ below wild-type levels in the absence of acetic acid stress. In cells grown in the presence of acetic acid, *tpk2*Δ reversed their expression in *tpk1*Δ mutant cells to the wild-type levels. Taken together, our data indicate that Tpk1 and Tpk3 play a redundant, negative regulatory role in the expression of *YRO2* and *YGP1,* while Tpk2 antagonizes the effect of Tpk1 and Tpk3. 

The opposing roles of the three catalytic subunits of PKA in a cellular process have been reported previously [28,45,46]. Tpk2 activates filamentous growth while Tpk1 and Tpk3 inhibit it. This is achieved through the interaction of the three catalytic subunits with different downstream effectors (reviewed in [47]). The mechanism behind the opposite effects of Tpk1 (Tpk3) and Tpk2 on the expression *YRO2* and *YGP1* is currently unknown.

### 3.6. haa1*Δ* Largely Abolishes the Increased Expression of YRO2, TPO2, TPO3, and SPI1 in tpk1*Δ* tpk3*Δ* Double Mutant Cells

A *tpk1*Δ *tpk3*Δ double mutation leads to the highest levels of expression of *YRO2–* and *YGP1–lacZ* reporter genes among *tpk* single and *tpk* double mutant cells grown in the absence of acetic acid treatment (Figure 4). To determine whether this is due to activation of Haa1, we introduced *haa1*Δ into *tpk1*Δ *tpk3*Δ double mutants and analyzed the expression of these reporter genes. We also included three more *lacZ* reporter genes under the control of the promoters of *TPO2*, *TPO3,* and *SPI1* in the analysis. Figure 5 shows that *haa1*Δ largely abolished the increased expression of *YRO2*, *TPO2*, *TPO3*, and *SPI1–lacZ* reporter genes due to *tpk1*Δ *tpk3*Δ in cells grown in the absence of acetic acid treatment. *haa1*Δ had little effect in reducing the expression of the *YGP1–lacZ* reporter gene in the *tpk1*Δ *tpk3*Δ double mutant. *haa1*Δ also had the weakest effect in reducing *YGP1–lacZ* expression compared to *YRO2–* and *TPO2–lacZ* reporter genes in *ras2*Δ mutant cells (Figure 2), suggesting that the increased expression of *YGP1–lacZ* due to *ras2*Δ and *tpk1*Δ *tpk3*Δ is dependent on additional transcription factor(s) and/or regulatory mechanisms. The reversal of the *tpk1*Δ *tpk3*Δ-induced expression of these reporter genes by *haa1*Δ suggests that *tpk1*Δ *tpk3*Δ *haa1*Δ triple mutant cells are sensitive to acetic acid stress, which is confirmed by the growth data in Figure 5F. Together, our data suggest that Haa1 is activated in *tpk1*Δ *tpk3*Δ double mutant cells, resulting in the increased expression of Haa1-dependent genes.
Figure 5Haa1 is largely required for increased expression of *YRO2–lacZ*, *TPO2–lacZ*, *TPO3–lacZ*, and *SPI1–lacZ* reporter genes in *tpk1/3*Δ mutant cells grown in MM4 medium. Wild type (BY4741) and isogenic *haa1*Δ (ZLY4043), *tpk1/3*Δ (ZLY4520), and *tpk1/3*Δ *haa1*Δ (ZLY5264) mutant strains carrying a centromeric plasmid encoding a *YRO2–lacZ* (panel (**A**)), *YGP1–lacZ* (panel (**B**)), *TPO2–lacZ* (panel (**C**)), *TPO3–lacZ* (panel (**D**)), or *SPI1–lacZ* (panel (**E**)) reporter gene were grown in MM4 medium without acetic acid to mid-logarithmic phase. β-galactosidase activity assays were conducted. The data are presented as the mean ± standard deviation. (**F**) *haa1*Δ single and *tpk1*Δ *tpk3*Δ *haa1*Δ triple mutant cells are equally sensitive to acetic acid stress. Wild type (BY4741) and isogenic *tpk1/2*Δ (ZLY4516), *tpk2/3*Δ (ZLY4522), *tpk1/3*Δ (ZLY4520), *haa1*Δ (ZDY166), and *tpk1/3*Δ *haa1*Δ (ZLY5264) mutant strains were serially diluted and spotted on MM4 medium without and with 60 mM acetic acid. “*” indicates a significant difference (*p* < 0.05) in the means of two groups of data indicated by the beginning and end of a horizontal line.
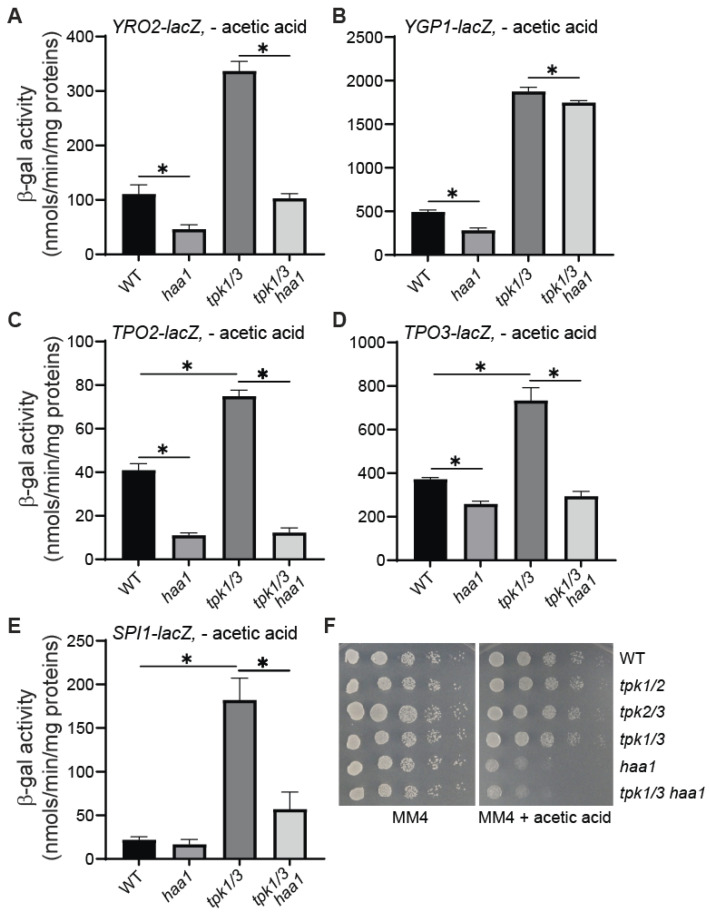


### 3.7. tpk1*Δ* tpk3*Δ* Double Mutants Yield Flocculent Cultures in Response to Acetic Acid Stress

When growing cultures for β-galactosidase activity assays, we noticed that the *tpk1*Δ *tpk3*Δ mutant cells are flocculent in the MM4 medium supplemented with 60 mM acetic acid (Figure 6A). In the MM4 medium without the supplementation of acetic acid, the *tpk1*Δ *tpk3*Δ mutant cells were not flocculent. The *tpk* single mutants, *tpk1*Δ *tpk2*Δ, and *tpk2*Δ *tpk3*Δ double mutant strains did not yield flocculent cultures in the MM4 medium supplemented with acetic acid either. Surprisingly, despite similar phenotypes exhibited by the *ras2*Δ and *tpk1*Δ *tpk3*Δ mutant cells, *ras2*Δ did not result in flocculation in the MM4 medium supplemented with 60 mM acetic acid. Culture flocculence is noticeable when large aggregates of cells are formed. To determine whether the *tpk1*Δ *tpk3*Δ double mutant cells in the absence of acetic acid treatment form small clusters, we examined cell cultures under microscope. The microscopic analysis showed that the *tpk1*Δ *tpk3*Δ mutant cells grown in the absence of 60 mM acetic acid had similar clustering (or lack of) to the wild-type, while they formed large clusters in response to acetic acid stress (Figure 6B,C). Since the *tpk1*Δ *tpk3*Δ mutant cells formed clusters only in the presence of acetic acid, we investigated whether Haa1 was required for the flocculation phenotype. We found that the *tpk1*Δ *tpk3*Δ *haa1*Δ triple mutant cells did not yield flocculent cultures in the MM4 medium supplemented with 60 mM acetic acid (Figure 6). The flocculation phenotype observed in Figure 6 may be related to the expression of *FLO11*. Mutations in *TPK1/3* and *TPK2* are known to have opposite effects on *FLO11* expression [45,46]. It is possible that the increased expression of *FLO11* in *tpk1*Δ *tpk3*Δ double mutant cells somehow leads to flocculation in response to acetic acid stress. Together, our data suggest that Haa1 activation in *tpk1*Δ *tpk3*Δ mutant cells grown in the presence of acetic acid contributes to culture flocculence. Flocculation can be beneficial in industrial fermentation processes by facilitating the removal of yeast cells (reviewed in [48]). This phenotype may be useful for engineering acetic acid-resistant strains for bioethanol production and other industrial fermentation processes.
Figure 6A *tpk1Δ tpk3Δ* double mutation leads to a Haa1-dependent flocculation phenotype in response to acetic acid stress. (**A**) Pictures of culture tubes containing wild type (BY4741) and isogenic *haa1*Δ (ZLY4043), *tpk1/3*Δ (ZLY4520), and *tpk1/3*Δ *haa1*Δ (ZLY5264) mutant strains grown in the MM4 medium without and with 60 mM acetic acid in the mid–late log phase. (**B**) Differential interference contrast (DIC) images of yeast cells from cultures, as described in panel (**A**). (**C**) A differential interference contrast image of a large aggregate of *tpk1/3*Δ cells (flattened between the cover slip and slide) grown in the MM4 medium supplemented with 60 mM acetic acid.
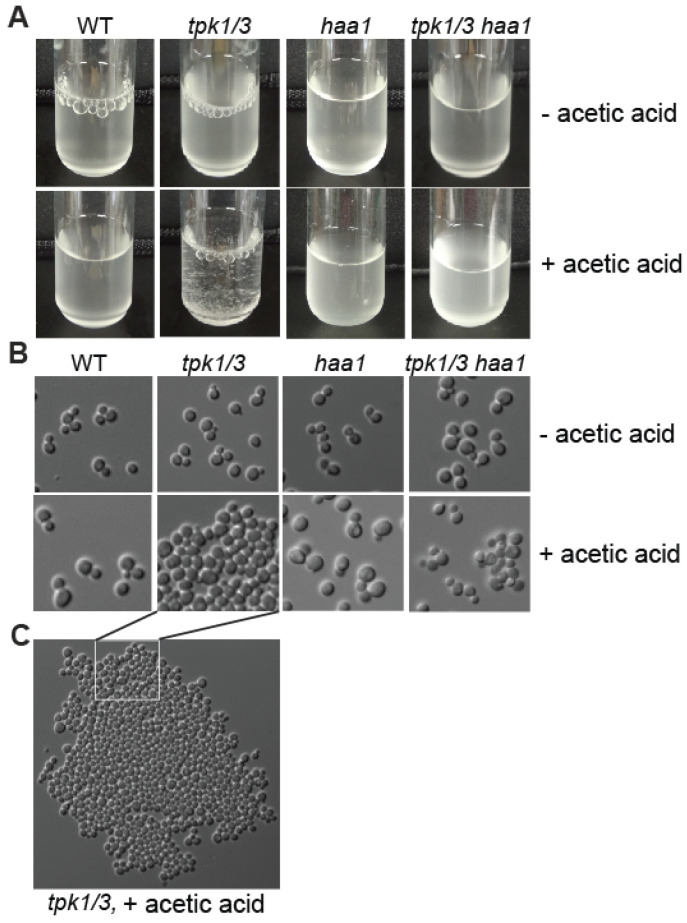


## 4. Discussion

Haa1 plays an important role in mediating cellular adaptation to acetic acid stress by activating the expression of many acetic acid-responsive genes. In this report, we show that PKA is a negative regulator of the acetic acid stress response. Our conclusion is based on the observation that PKA-activating mutations, namely, *pde2*Δ and *RAS2A18V19*, reduce the expression of Haa1-regulated genes. Conversely, mutations that reduce the activity of PKA, *ras2*Δ*,* and *tpk1*Δ *tpk3*Δ increase the expression of Haa1-regulated genes. The reduced expression of Haa1-regulated genes in *pde2*Δ and *RAS2A18V19 * mutants correlates with increased sensitivity to acetic acid. In contrast, the increased expression of Haa1-regulated genes in *ras2*Δ mutants improves cellular fitness in response to acetic acid stress. PKA is an important regulator of cell growth, development, and stress response pathways. Our report cements a new role for PKA as an important player in the acetic acid stress response.

How do PKA and Haa1 work together in mediating the expression of Haa1-responsive genes? One possibility is that Haa1 acts downstream of PKA in a linear pathway. In this scenario, PKA negatively regulates the activity of Haa1. Alternatively, PKA and Haa1 may work in parallel pathways in an antagonistic manner. We failed to detect an interaction between Haa1 and Tpk1, Tpk2, or Tpk3 via yeast two-hybrid and/or coimmunoprecipitation analyses, as well as significant changes in the phosphorylation state of Haa1 in mutants that alter the PKA activity. Although it can be challenging to observe an interaction between a protein kinase and its target due to the inherent disengagement after the target is phosphorylated, the failure to detect an interaction between Haa1 and Tpk1/2/3 increases the possibility that PKA may regulate Haa1 indirectly or function in a parallel pathway of Haa1. Other regulatory factors, for example, Msn2/4 and Yak1, have been associated with the expression of acetic acid-responsive genes [1,16,28]. Msn2/4 and Yak1 have been proposed to function downstream or in a parallel pathway of PKA [21,28,49,50]. Future work is needed to elucidate the mechanism by which PKA regulates the expression of Haa1-dependent genes.

PKA regulates the expression of acetic acid-responsive genes, so it is conceivable that PKA activity may be regulated by acetic acid. If so, how does PKA sense acetic acid? PKA is under the control of many positive and negative regulators. There can be many possibilities. One facile explanation is that reduced cytoplasmic pH in response to acetic acid treatment regulates the activity of PKA. Cytoplasmic pH has been proposed to be a second messenger that regulates the PKA pathway [51]. In their study, Dechant et al. proposed that high cytoplasmic pH activates PKA while low pH inactivates it. Incubation of yeast cells with acetic acid in the growth medium at a pH below the pK_a_ value of acetic acid can lead to a reduction in the cellular pH (reviewed in [1]). We propose that low cytoplasmic pH due to acetic acid stress leads to reduced PKA activity and consequently increased expression of Haa1-regulated genes. However, this notion seems to contradict the observation of PKA activation by low pH [52]. In their study, Colombo et al. found that the addition of the protonophore 2,4-dinitrophenol to yeast cells at low extracellular pH caused an activation of Ras2 and an increase in cAMP levels. Future work is needed to reconcile these discrepancies.

The involvement of PKA in the acetic acid stress response in yeast opens new possibilities for designing acetic acid-resistant strains for biofermentation applications. Research exploring genetic and biochemical strategies to improve yeast robustness against acetic acid stress can ultimately enhance the efficiency and productivity of fermentation-based industries.

## Data Availability

The original contributions presented in the study are included in the article, further inquiries can be directed to the corresponding author.

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
