# Peer review of "Protein Kinase A Negatively Regulates the Acetic Acid Stress Response in S. cerevisiae"

_microorganisms, 2024, doi:10.3390/microorganisms12071452_

Round 1

Reviewer 1 Report

Comments and Suggestions for Authors

The paper  of Bourgeois et al showed that PKA could be  a negative regulator of the acetic
 acid stress response in budding yeast likely acting on the Haa1 transcription factor.

The results clearly indicate that Ras2  and Pde2  activities are relevant for the regulation of Haa1 suggesting that a decrease of PKA activity (given by Ras2 deletion) should activate Haa1 dependent transcription, while an increase given by the expression of RAS2Val19 allele or by PDE2 deletion had the opposite effect. However the results obtained by deletion of TPK1,2,3 genes are conflicting with an opposite effect given by deletion of tpk1 and/or tpk3 in comparison with tpk2 deletion, and this is difficult to explain since all the three genes encodes for a PKA catalytic subunit and should be discussed better. IN addition the authors generally speak of "stress" without mentioning that acetic acid induces cell death and apoptosis in yeast and there are evidences that PKA is involved in these processes (see FEBS Lett 597 (2023) 298–308     and Front CellDev Biol 9:642375). These facts should be mentioned in the introduction were there are instead 21 references !!... (that should be reduced in my opinion) for stress induced by acetic acid (lines 55-62 of the manuscript). In my opinion the manuscript should be partially rewritten at least in the Introduction and in the Discussion and additional informations should be given in the Mat and Meth section.
Minor questions:
1) References should be  numbered in order of citation.
2) Figures are not in the proper order ( start with Fig4 ?!!)

Specific points:

 Line 55-62  too many references (21) only for acetic acid stress, not always pertinent and with evident bias (no mention on the cell death and apoptosis induced by acetic acid..)

Line 100-102 this  sentence  is not true , there as some informations on this point see for example  Cell Cycle 8, 1256 (2009) and Amigoni et al Oxid Med Cell Long 2013: 678473 or Front CellDev Biol 9:642375 (2021) . Ref 53 and Ref 40 are not pertinent here.

Line 110 here should be added Ref  40 (Salas et al 2022), while reference Toda et al should be withdrawn

Lines 113-114 Reference Salas et al 2022 should be withdrawn here while papers specific for Gpa2 should be cited, for example Colombo et al EMBO J. 17, 3326 (1998)).

Line 129 the supplement for auxotrophic requirements should be added also in MM4 medium

Line 137 pRS303-RAS2A18V  give a reference for this plasmid

Line 147  by tetrad analysis ?? is not clear what means, did you used a heterozygous diploid for sporulation? why? please explain

Table 1  SGDP please give a proper reference

Line 205 YCp50-RAS2A10V19  give a reference , the same in the Table2 - Kelly Tatchell, ref....

Line 280 is not clear how were generated the deletion mutants cited here

Fig.3  panel B  the mutants haa1 and pde2 start growth after 20 hrs, is acetic acid still present or has been metabolized at that point?

A question: what is the growth kinetics of the tpk1, tpk2 and tpk3 mutants? This experiment could be usefull

In Discusion Line 623 this point should be discussed better since has been reported that acidification (for example done with Dinitrophenol...) caused an activation of Ras2 and therefore an increase of cAMP and of PKA activity (Colombo et al. EMBO J 17, 3326 (1998).

Reviewer 2 Report

Comments and Suggestions for Authors

Review

for the article entitled Protein kinase A negatively regulates the acetic acid stress response in S. cerevisiae"

It is regrettable that I am unable to assess the article "Protein kinase A negatively regulates the acetic acid stress response in S. cerevisiae" for potential publication in Microorganisms. This is due to the fact that the submitted version of the article gives rise to confusion with regard to the presentation of results in figures, tables and their descriptions. To illustrate,

Point 1: pages 5-6 – The rationale for the presentation of the figures describing the properties of the mutants in the Materials and Methods section is not readily apparent. It is recommended that the figures and their descriptions be relocated to the Results section.

Point 2: page 9, lines 267–302, it is unclear where Figure 1 is located, which should show the effect of ras2∆ and pde2∆ on the expression of the YRO2 and YGP1-lacZ reporter genes.

The authors are advised to  include  the figures and tables directly under the description of the results; the figures and tables must be consecutively numbered.

Point 3: In the Introduction and other sections, reference numbers should be placed in square brackets [ ], for example [1], [1–3] or [1,3].

Round 2

Reviewer 1 Report

Comments and Suggestions for Authors

The paper has been revised properly and the authors responded in a positive way to all the raised questions. The reference list has been modifyed as required and also the Methods have been described better and with adequate details

Reviewer 2 Report

Comments and Suggestions for Authors

The report can be found in the attached file.
